# *Aurora Borealis* (Bora), Which Promotes Plk1 Activation by Aurora A, Has an Oncogenic Role in Ovarian Cancer

**DOI:** 10.3390/cancers12040886

**Published:** 2020-04-06

**Authors:** Alfonso Parrilla, Marta Barber, Blanca Majem, Josep Castellví, Juan Morote, José Luis Sánchez, Asunción Pérez-Benavente, Miguel F. Segura, Antonio Gil-Moreno, Anna Santamaria

**Affiliations:** 1Group of Biomedical Research in Urology, Cell Cycle and Cancer Laboratory, Vall Hebron Research Institute (VHIR), Universitat Autònoma de Barcelona (UAB), 08035 Barcelona, Spain; alfonso.parrilla@vhir.org (A.P.); marta.barber@vhir.org (M.B.); blanca.majem@vhir.org (B.M.); jmorote@vhebron.net (J.M.); 2Department of Pathology, Vall Hebron University Hospital, 08035 Barcelona, Spain; jcastellvi@vhebron.net; 3Department of Urology, Vall Hebron University Hospital, 08035 Barcelona, Spain; 4Group of Biomedical Research in Gynecology, Vall Hebron Research Institute (VHIR), Universitat Autònoma de Barcelona (UAB), CIBERONC, 08035 Barcelona, Spain; jlsanig@yahoo.es (J.L.S.); twinsma@me.com (A.P.-B.); antonioimma@yahoo.es (A.G.-M.); 5Department of Gynecology, Vall Hebron University Hospital, 08035 Barcelona, Spain; 6Group of Translational Research in Child and Adolescent Cancer, Vall Hebron Research Institute (VHIR), Universitat Autònoma de Barcelona (UAB), 08035 Barcelona, Spain; miguel.segura@vhir.org

**Keywords:** PLK1, BORA, ovarian cancer, mitosis, oncogene, targeted therapy

## Abstract

Identifying novel actionable factors that critically contribute to tumorigenesis is essential in ovarian cancer, an aggressive and disseminative tumor, with limited therapeutic options available. Here we show that *Aurora Borealis* (BORA), a mitotic protein that plays a key role in activating the master mitotic kinase polo-like kinase 1 (PLK1), has an oncogenic role in ovarian cancer. Gain and loss of function assays on mouse models and ex vivo patient-derived ascites cultures revealed an oncogenic role of BORA in tumor development and a transcriptome-analysis in clinically representative models depicted BORA’s role in survival, dissemination and inflammatory cancer related-pathways. Importantly, combinatory treatments of FDA-approved inhibitors against oncogenic downstream effectors of BORA displayed synergistic effect in ovarian cancer models, offering promising therapeutic value. Altogether, our findings uncovered for the first time a critical role of BORA in the viability of human cancer cells providing potential novel therapeutic opportunities for ovarian cancer management.

## 1. Introduction

Turning on polo-like kinase 1 (PLK1) kinase is the responsibility of BORA (*Aurora Borealis*). BORA, first identified in *Drosophila melanogaster* as a co-activator of AURORA A [1], controls the proper timing of mitosis onset by activating PLK1 [2,3] in human cells. Similarly, in conditions of DNA damage and activation of the G2/M checkpoint, ataxia telangiectasia and Rad3 related kinase (ATR) phosphorylates BORA to degrade it and to sustain the G2/M blockade [4]. Under recovery conditions, Cyclin-dependent kinase 1 (CDK1)-mediated phosphorylation of BORA on its N-terminal domain is essential for PLK1 re-activation and thus mitotic commitment [5,6,7], underlying its crucial role in cell cycle division. Nevertheless, even though BORA depletion has been reported to reduce the activity of PLK1 kinase, our understanding of its relevance in cancer is still very limited and there is no comprehensive study that defines the downstream biological consequences of BORA modulation in cancer. Recent evidence has shown that BORA is overexpressed in various tumors such breast, lung, and gastric adenocarcinomas and might serve as a prognostic biomarker [8]. 

Ovarian cancer (OC), the most lethal gynecologic malignancy, is frequently diagnosed at advanced stages with disseminated disease, which limits the therapeutic options [9]. Despite improved cytoreductive surgical approaches and chemotherapy-based regimens, the survival of OC patients has remained largely unchanged during the last two decades [10,11]. Recent advances in cancer genomics have revealed that OC is molecularly a very heterogeneous disease, with extensive genomic instability, copy-number variations and defects in the homologous recombination repair pathway [12]. These genomic aberrations contribute to the development of tumor resistance, hampering effective treatment and ultimately causing disease recurrence [13], but also offer novel potential actionable vulnerabilities that can enhance the effectiveness of existing therapies.

Molecular targeted therapies have been implemented routinely into the clinics changing OC management with the inclusion of anti-angiogenic compounds (i.e., monoclonal antibodies such Bevacizumab) [14,15] and poly ADP-ribose polymerases (PARP) inhibitors (i.e., Olaparib or Rucaparib) for breast-cancer (BRCA)-mutated carriers and BRCAness phenotype patients [16,17]. Synthetic lethality produced by PARP inhibitors enhances the therapeutic window after chemotherapy in recurrent platinum-sensitive OC subjects [18]. Nonetheless, it is effective in only a subset of patients, highlighting the urgent clinical need of searching new therapeutic approaches for a larger number of OC patients.

While the first generation of antimitotic drugs aimed at blocking cell division (classical antimicrotubule agents), the new generation is exploiting novel cancer-specific vulnerabilities such as the generated chromosomal instability (CIN) [19]. CIN-inducing cancer therapies target mitotic-specific alterations such as centrosome amplification or overexpressed checkpoint regulators to adversely impact in chromosome segregation, triggering cell death and thus trying to maximize clinical results [20,21]. Some of them, focused on the G2/M DNA damage checkpoint (e.g., PLK1, WEE1 G2 checkpoint kinase (WEE1) or telangiectasia mutated kinase (ATM)) are being investigated clinically in many cancers with promising results [22,23,24]. Volasertib (BI-6727), an ATP-competitive PLK1 inhibitor, vastly explored preclinically [25], has achieved clinical benefits in OC [26] and it received the breakthrough therapy designation by the US Food and Drug Administration (FDA) due to its substantial therapeutic effect in patients with acute myeloid leukemia [27]. However, its nonspecificity can cause undesirable adverse effects, which lead to reconsider its use as a clinical agent. In this regard, understanding the role of BORA in cancer cells will add valuable insights into BORA/PLK1-related mechanisms and might offer novel opportunities for therapeutic intervention in OC.

Here, we mined through public transcriptome datasets to identify cell cycle-related genes that could be contributing to the aggressive behavior of OC and we found BORA to be highly expressed in a myriad of OC tissue specimens compared to benign samples and a correlation with poor overall survival. We also have shown that ectopic expression of BORA is associated with malignant transformation features in vitro and fosters tumorigenesis in vivo. Moreover, knocking down BORA impairs OC viability in vitro, in vivo and ex vivo in patient-derived ascites cells grown in anchorage-independent conditions. In agreement with these results, the transcriptome analysis of BORA-depleted cells revealed modulated genes involved in survival, dissemination and inflammation-related pathways. Furthermore, combined inhibition of BORA downstream effectors (i.e., B-cell lymphoma 2 (BCL2) and Cyclin-dependent kinase-6 (CDK6) showed remarkable therapeutic potential in OC models. Collectively, these results define a novel oncogenic role of BORA and open up potential new therapeutic avenues for OC management.

## 2. Results

### 2.1. An Integrated Bioinformatics Screening Identifies Mitotic Regulators Potentially Involved in OC

In an attempt to identify therapeutically actionable candidates in OC, we performed an integrative computational analysis of transcriptomics data combined with survival outcome using different cohorts of OC specimens. First, we analyzed the GSE14407 dataset, which includes comprehensive genomic information of healthy ovarian surface epithelial samples (*n* = 12) compared to serous OC epithelial tissues (*n* = 12). A total of 2280 genes were found differentially expressed between these two groups (Fold change > ± 2.5 and FDR = 0.01; Figure 1A). Most genes were linked to relevant cancer-related functions including cell division, p53 and PI3K-AKT signaling pathways (Appendix A). Focusing on the upregulated genes, a Gene Ontology (GO) functional term enrichment identified 10 clusters (*p*-value < 1.2x10^−4^; Figure 1B) with cell and mitotic division being the top significant enriched deregulated processes. Taking into consideration that the mitotic spindle apparatus encompasses a plethora of validated targets currently used as standard-of care chemotherapies for the treatment of multiple cancers [28,29], we searched for unexplored hits within these top deregulated GO terms. Candidates selected for functional studies included those genes that (i) were listed in the mitosis and cell division GO terms; (ii) whose higher expression was associated with worse overall patient survival (*p*-value < 0.05) and (iii) that had not been previously described in OC tumorigenesis. As summarized in Appendix A, *BORA*, *SPC25*, *SPC24*, *KIF20B*, *FAM64*, *CCNA*, *CDC5A* and *OPI5* were the genes with higher score (Figure 1C). We and others have previously shown that CDK1-dependent BORA phosphorylation is crucial for PLK1 activation in G2/M checkpoint recovery, where PLK1 is indispensable to re-enter into mitosis [30], supporting a key role of BORA in cell progression. Moreover, BORA has been recently shown to be upregulated in other tumor types [8] and as a biomarker of radiation response in human lymphoblastoid cell lines [31], but little is known about the possible functional implication of BORA itself in cancer. Therefore, we selected BORA for further studies.

### 2.2. BORA Overexpression Is Associated with Poor Prognosis in OC Patients

We first validated that BORA levels were consistently upregulated in ovarian tumoral tissues compared to normal tissue samples in four additional OC data sets (Figure 1D). Furthermore, analyses based on The Cancer Genome Atlas (TCGA) dataset, which includes genomic information of 541 OC patients, showed that BORA expression correlated with high neoplasm histological grade and advanced clinical stage of the disease (Figure 1E,F). In agreement with the already established function of BORA in mitotic division, we also found a positive correlation of BORA levels with the proliferation marker Ki67 (Figure 1G), suggesting that expression of BORA labels highly proliferative tumor cells. Moreover, expression of BORA also showed positive correlation with the CIN25 score, a measure of chromosome instability linked to tumor aggressiveness [32] (Figure 1H). In contrast, no correlation was observed with MUC16 expression (CA-125 antigen), a routinely used biomarker of OC recurrence (Appendix A). Analyses of the different TCGA publicly-available data sets for all major cancer types revealed that BORA levels were also increased in other tumor types, when compared to a dataset that comprises the expression levels of normal tissues (GSE3526), particularly in head and neck, leukemia and colon carcinomas (Figure 1I). Moreover, high BORA levels correlated with worse overall survival in breast, lung and liver tumors (Appendix A). Of note, genomic alterations in BORA locus were rather low in all types of cancer (Appendix A) without any particular hotspot mutation across the length of the protein (Appendix A).

We next validated these results in an independent cohort of 40 ovarian human tissue samples obtained from the Vall d’Hebron Hospital (Barcelona, Spain). Higher BORA mRNA expression was seen in tumor primary tissues compared to benign ovarian samples (Figure 1J), and a significant correlation was observed with advanced stage (Figure 1K). A positive tendency, albeit non-significant, was observed in undifferentiated grade tumors (Appendix A) and no differences in BORA expression were observed among different histological subtypes of OC (Appendix A). In addition, the levels of PLK1 were also analyzed in this cohort, finding that PLK1 levels were also increased in the tumoral tissues compared to benign samples (Appendix A). We further analyzed the mRNA levels of BORA in 13 paired tumor-metastasis patient-samples (26 samples in total) and a significant upregulation of BORA was seen in metastasis compared to matched primary tumors (Appendix A), strengthening the notion that BORA levels correlate with OC aggressiveness. At protein level, mild or not-detectable BORA levels were found in the majority of benign samples, in contrast to the consistent expression detected in all high-grade serous carcinoma (HGSC) tissue specimens tested. PLK1 protein expression and its activity levels through a subrogated marker, the phosphorylation in pTCTP-Ser46 [33], displayed also an increase in HGSC samples compared to the benign ovaries (Figure 1L). Concomitant with these results, BORA mRNA and protein levels were overexpressed in a panel of OC cell lines when compared to the nontumorigenic immortalized ovarian surface epithelium (IOSE) cell line (immortalized by SV40 T/t) that displays almost undetectable endogenous protein levels of BORA. No obvious correlation with the histological tumor type or p53 status was observed (Figure 1M and Appendix A). A significant correlation between BORA mRNA and protein levels was observed in OC cell lines (Appendix A). Taken together, these results demonstrate that BORA is overexpressed in OC and might be an indicator of adverse patient prognosis.

### 2.3. BORA Overexpression Renders Malignant Transformation Of Nontumoral Cells In Vitro

The fact that BORA is aberrantly upregulated in OC specimens prompted us to speculate that BORA could be a potential causal factor promoting ovarian tumorigenesis. To test this hypothesis, we engineered an IOSE cell line containing the human coding sequence of BORA (pIND_BORA) to perform gain-of-function studies (Figure 2A). We examined the IOSE-pIND_BORA cells’ capability of anchorage-independent growth in soft agar, a classical hallmark of cellular malignant transformation in vitro [34]. We found that BORA overexpressing cells doubled the number of colonies formed after three weeks of anchorage-independent growth in soft agar (Figure 2B). Concomitantly, BORA-overexpression was accompanied by a higher proliferation rate in 2D (Figure 2C) and accelerated the migration capacity of IOSE cells by twofold compared to IOSE cells infected with the empty vector (pIND_EV) (Figure 2D). Moreover, when cells were left to reach confluence, IOSE-pIND_BORA cells kept growing, with cells forming multiple layers; however, control cells died after inhibition of growth by cell contact (Figure 2E). The result was consistent with a nesh-network in BORA overexpressing cells, promoting the loss of growth by contact inhibition in IOSE cells, another classical hallmark of tumorigenicity [35]. These results indicate that BORA overexpression induces the malignant transformation of IOSE cells in vitro. We also ectopically increased BORA expression in the tumoral SK-OV-3 cell line, which present low-medium endogenous levels of BORA (Appendix A). BORA-overexpressing cells also showed increased proliferation (Appendix A) and higher capacity to form colonies in soft agar (Appendix A), thereby suggesting that high levels of BORA can contribute to tumor progression in transformed cells.

### 2.4. BORA Contributes to Ovarian Tumorigenesis In Vivo

To go a step further in the oncogenic transformation role of BORA, we assessed in vivo tumorigenicity of BORA-overexpressing ovarian epithelial cells. Firstly, IOSE-pIND_EV and pIND_BORA cells were injected subcutaneously into the flank of nude mice. Although after three weeks post-injection we observed an attempt of tissue engraftment in more than 40% of the mice (Appendix A), eventually none of the mice with IOSE implanted cells developed a tumor. Histopathological analysis revealed neither malignant cells nor preneoplastic tissue in any of the mice, although they exhibited single tiny scars, presumably remnants of an initial intention of tumor engraftment. These scars appeared as small fibrotic lesions together with signs of chronic inflammation and fat tissue (data not shown). We then analyzed the effects of overexpressing BORA in the SK-OV-3 cell line, when xenografted in NMRI nude mice. One-week post-injection, all mice bearing BORA overexpressing SK-OV-3 cells engrafted and developed tumors, while in the control group (SK-OV-3 pIND-EV) none did so until week 8 (Figure 2F). Moreover, SK-OV-3-BORA-overexpressing tumors displayed, not only faster, but increased growth compared to the control group (Figure 2G), with an average size of 2089 ± 444 mm^3^ in contrast to SK-OV-3-EV tumors that measured 1209 ± 316 mm^3^ 26 days post-engraftment. Immunoblot analysis of the resected tumors verified the overexpression of BORA (Figure 2H) and histopathological analysis revealed an increase in the number of Ki67-proliferating cells in these tumors (*p* < 0.05; Figure 2I). Taken together, BORA clearly contributes to OC tumor growth in vivo.

### 2.5. BORA Silencing Has Tumor-Suppressive Effects In Vitro

The function of PLK1 in sustaining the proliferative capacities of cancer cells has been previously reported [36]. Thus, having established BORA as a determinant factor for ovarian tumorigenesis, we predicted that BORA might also regulate the viability of OC cells. Knocking down BORA with two independent lentivirus-based shRNAs reduced protein levels efficiently in all tested ovarian cell lines (Figure 3A). A significant impairment of cell proliferation was observed in the five OC cell lines studied, especially with shBORA#2 (hereinafter shBORA); while the effect in the IOSE line was considerably less pronounced (Figure 3B). We then proceeded to deepen into the phenotypical consequences of BORA depletion. Fluorescence-activated cell sorter FACS analysis revealed that BORA silencing promoted a G2/M phase arrest in SK-OV-3 and A2780p cells with a concomitant decrease in the percentage of cells in G1 phase (*p* < 0.05; Figure 3C), concurring with previous reports [3,37], while IOSE shBORA -infected cells displayed a similar cell cycle prolife to shCTL cells (Figure 3C). Moreover, chromatin staining of shBORA-infected cells showed an apoptosis-consistent pattern of condensed and/or fragmented chromatin 96 h post-transduction, while shCTL -infected cells displayed uniform chromatin staining (white arrowheads; Figure 3D; data quantification in Appendix A). BORA knockdown was accompanied by a reduction in the surrogate marker of PLK1 activity, pTCTP-Ser46, but not in the basal levels of PLK1 and AURORA-A kinases as previously reported [3], suggesting that, at least, part of the consequences of BORA depletion are due to reduced PLK1 activity (Figure 3E). Cyclin B1 immunoblot in SK-OV-3-transduced shBORA cells confirmed the pronounced G2/M arrest followed by higher PARP cleaved levels pointing towards a caspase-dependent apoptosis after BORA depletion in tumoral cells (Figure 3E). BORA depletion was also accompanied by lower migration capacities in transwell assays (Figure 3F). To extend our findings of BORA functions in other tumors, we silenced BORA in a subset of other cancer cell lines from prostate, endometrial, neuroblastoma and colon cancer. We observed that BORA depletion resulted in decreased cellular proliferation in all models tested (Appendix A). Long-term colony formation assays also reflected a reduced number of colonies, when transduced with shBORA viruses, compared to shCTL cells (Figure 3G,H). Interestingly, all BORA-depleted surviving colonies in the A2780p cell line expressed lower, but residual levels of BORA compared to clones derived from A2780 shCTL (Appendix A), thereby suggesting that minimal BORA expression is able to sustain cellular division, in agreement with the report of Bruinsma et al. [38]. To ascertain if OC cells were able to survive with complete BORA depletion, we designed a gRNA to target exon 2 of BORA and cloned it into a CRISPR/Cas9 vector (pSpCas9 (BB)-2A-GFP) [39]. We failed to obtain single cell clones with no BORA protein levels at all, and only clones with reduced but residual BORA levels were obtained, which mirrored our previous results with shRNA-mediated depletion of BORA (Appendix A). These data support the notion that the complete depletion of BORA is most likely lethal for cells, but much reduced levels of BORA are enough to sustain cellular proliferation of OC cells.

### 2.6. BORA Depletion Impairs Tumor Growth In Vivo

We next analyzed the effects of silencing BORA in an OC subcutaneous xenograft model. SK-OV-3 cell line transduced with constitutive shCTL or shBORA lentiviral particles for 48 hours were injected in the flank of NMRI nude mice. BORA depletion was monitored with a fraction of infected cells by immunoblot (Appendix A). At six weeks post-injection, 85% of mice bearing shCTL cells developed tumors compared to only 42% of mice bearing shBORA-infected cells did. The last BORA-depleted tumor eventually engrafted at 22 weeks post injection (Appendix A; *p*-value < 0.05). Moreover, tumors detected in the shBORA group progressed at a very low pace thorough the experiment (Appendix A). At end-point, excised tumors from the shCTL were larger and heavier compared to BORA depleted tumors (Appendix A). Immunoblot using protein lysates from representative tumors confirmed the downregulation of BORA in shBORA-bearing tumors accompanied with an increase in apoptotic markers (Appendix A). Together, constitutive BORA depletion in SK-OV-3-implanted mice, impacts on tumor engraftment, modulating tumor fate and thereby reducing tumor growth.

To mimic a potential therapeutic intervention in the clinics, we proceeded to engineer an inducible vector to modulate BORA expression in vivo once the tumor is formed. We used the Tet-On lentiviral inducible vector pTRIPZ (Dharmacon), which provides inducible depletion of BORA in the presence of doxycycline. SK-OV-3 cells were infected with pTRIPZ-EV (empty vector) and two vectors that target BORA (harboring different shRNA sequences); pTRIPZ-BORA-1 and pTRIPZ-BORA-2. The doxycycline addition to the culture media successfully depleted BORA in the cells transduced with either of the two vectors compared to empty vector (Appendix A). The inducible-mediated BORA silencing resulted in a reduction of the proliferation and colony-formation capacities, while the doxycycline alone did not have any affect (Appendix A), mirroring our previous observations. Next, we established subcutaneous xenografts tumors by the injection of the SK-OV-3 cells transduced with pTRIPZ-BORA-1 cells. When tumors reached a volume of ~150 mm^3^ on average, mice were randomized into two experimental groups: the BORA depletion group, which received doxycycline (1 mg/mL) together with 2% of sucrose in drinking water *ad libitum* and the control group treated with vehicle (2% sucrose). Differences in tumor growth between control and BORA inducible depletion were significantly visible one week after doxycycline administration and maintained throughout to the end of the experiment (Figure 4A). Animal weight was not different between the two groups through the experiment (Appendix A). At the endpoint, mice treated with sustained doxycycline showed a significant reduction in tumor size and weight (Figure 4B,C) and a positive correlation was observed between tumor volume and weight in BORA-depleted tumors (Appendix A), indicative of the accuracy of the measurements. BORA and tRFP expression were verified by immunoblot in these tumors, confirming that BORA was consistently depleted (Figure 4D). Histopathological analyses revealed that BORA-depleted tumors displayed reduced cellularity with extracellular matrix fueling empty spaces left by death cells, and a significant reduction in the percentage of Ki67 positive cells compared to untreated tumors (*p* < 0.001; Figure 4E). Collectively, these results suggest that inhibition of BORA might be a valuable therapeutic approach in OC.

### 2.7. BORA Silencing Reduces the Number and Viability of Patient-Derived Ascites Cells

Tumor ascites cells, which survive aggregated into multicellular spheroids in the peritoneal cavity, represent a hallmark of OC [40]. These cells are chemoresistant and responsible for driving OC progression at later stages and eventually causing disease recurrence [41]. Thus, to further evaluate the potential clinical implications of our findings in advanced OC, we established a preclinical ex vivo model using the ascitic fluid from three metastatic OC patients collected at the time of surgery. Tumoral patient-derived ascitic cells were cultured and expanded as previously described [42,43]. Then, these cells were reversely transduced with shCTL and shBORA lentiviral particles and cultured as spheroids in anchorage-independent conditions. At 36 h post transduction, the number of spheroids was scored. A significant reduction in the size and number of spheroids in two out of three patients was observed upon BORA depletion (Figure 4F,G). Immunoblot analysis confirmed the reduction in BORA levels in the three patients (Figure 4I). Furthermore, a reduction in cell viability was observed, remarkably in VH-3 patient, where the viability decreased up to 50%. (Figure 4H). Caspase-dependent apoptosis was observed in the three patients concurring with our previous results (Figure 4I). Demonstrating that the effects of BORA depletion are reproducible in patient-derived models grown in 3D and highlights BORA as a valid future therapeutic target for aggressive and disseminated ovarian tumors.

### 2.8. Reduction in BORA Levels Impact in Multiple Cancer-Related Genes

Having established that BORA could represent a promising therapeutic agent, we wanted to discern the landscape of proteins and pathways modulated upon BORA silencing that caused this phenotype and thus identify potential BORA target genes hitherto unknown. To this end, a whole-genome expression analysis in SK-OV-3 cell line at 48 h was performed in BORA-depleted and control cells (Appendix A). Principal component analysis segregated samples on the basis of shCTL versus shBORA (*n* = 3/group), indicating a robust and consistent transcriptional impact of BORA silencing (Figure 5A). After BORA knockdown, 192 genes were found to be upregulated whereas 215 were downregulated (Fold change >± 1,5; *p*-value <0.05; Figure 5B). The top differentially expressed genes with higher fold change variation are listed in Appendix A. The graph in Figure 5C summarizes the results from a gene set enrichment analysis (GSEA) of the transcriptome from BORA depleted cells. We observed enrichment in functions related to tumor biology; including cell commitment, disseminative process and inflammatory response (Figure 5C). Furthermore, gene sets composed of genes involved in energy production and cardiovascular system were found to be negatively enriched in shBORA cells, thereby indicating impairment in these processes at the transcriptional level upon BORA knockdown (Appendix A). Heatmaps and enrichment plots depicting the most relevant deregulated pathways illustrated how BORA knockdown impacts on the expression of several key oncogenic genes, such as *BCL2, CDK6, MMP7* or *NF-kB* signaling factors (such as *RELA)* (Figure 5D,E). Additionally, we selected a variety of cancer related candidates from the list of genes differentially expressed and validated their expression levels by RT-qPCR (Figure 5F). Interestingly, BORA depletion resulted in a significant downregulation of genes involved in cell proliferation (*BCL2*, *CDK6*, *RERG*), migration (*MMP7*) and cell cycle (*MARK2*, *CLASP2*). It also showed downregulation of the putative tumor-suppressor *SFRP1* and other genes involved in metabolic pathways (*SLC25A10*) and upregulation of the cytokine *IL1B*, the protein involved in the contractile system of striated and smooth muscles *TPM1*, and other genes with uncertain roles in cancer (*SHROOM2* and *RHOB*). We explored whether these proteins were modulated in two patient-derived tumoral cells and showed that the mRNA levels of these genes were also diminished upon BORA depletion (Figure 5G). To confirm the deregulated transcriptomic pathways obtained, BORA depleted and control mice tumors were used in an immunoblot analysis. We showed a reduction in key cancer related proteins: BCL2, CDK6, p65 and JNK1 kinase (Figure 5H), suggesting that reduction of the expression of these proteins might contribute to the effects of BORA depletion on cell survival. 

### 2.9. CDK6 and BCL2 Inhibitors Exert Synergistic Effect on OC Viability

Pharmacological inhibition of BORA is not possible. To date, no BORA inhibitors have been developed due to the unstructured nature of the protein. Nonetheless, since the levels of CDK6, and the prosurvival protein BCL2 were modulated upon BORA silencing and inhibitors of these pathways are FDA-approved drugs for breast cancer and leukemia [44,45], we sought to ascertain whether the combination of these inhibitors could mimic the BORA depletion phenotype and thus could result in beneficial therapeutic effects for OC patients. To test this, we used two CDK6 inhibitors, Palbociclib and Abemaciclib, and two BCL2 inhibitors, Navitoclax and Venetoclax; alone or in combination. The inhibition of both pathways individually resulted in a loss of proliferation in SK-OV-3 and A2780 cells (Appendix A). Notably, the combination of Palbociclib and Navitoclax showed strong synergism (CI value < 1) when the two drugs were combined at different concentrations in two cell lines (Figure 6A,B and Appendix A). To corroborate these results ex vivo, we examined the responses of two advanced OC patients to the combinatory treatment mentioned above (Appendix A). Once spheroids were formed, the combination drugs attenuated both sphere size and viability capacities (Figure 6C,D). Immunoblot confirmed an increase of active cleaved caspase-3 form and processing of PARP when both drugs were combined in the two patients (Figure 6E). Our preclinical results let us propose that the combined inhibition of BORA downstream effectors CDK6 and BCL2, could represent a promising therapy for OC. 

## 3. Discussion

Clinical management of OC remains a challenge due to its complex and instable intrinsic genetic nature. Antimitotic-based therapy (i.e., taxanes derivatives) represents the most widely used standard of care for cancer treatment, ultimately causing cell death by triggering extensive chromosome missegregation. Previous studies showed that BORA-knockdown cells tend to display unaligned chromosomes reducing the inter-kinetochore tension and leading to the activation of the spindle checkpoint, hence fulfilling important functions in spindle assembly [3,37]. In the present study, we have uncovered the biological consequences of BORA depletion as well as hitherto unanticipated pro-oncogenic functions in OC, making it a potential target for therapeutic intervention. 

Our data suggest that BORA may have prognostic value in ovarian tumors. Overexpression of BORA at protein level has been recently explored in several tumor types [8], but not yet in OC. MRNA and protein expression analyses indicate that high levels of BORA are increased in tumoral compared to benign samples and it correlates with tumor aggressiveness. These findings hold significant clinical application for BORA as biomarker in the future, to predict survival outcome and to identify high-risk OC patients to stratify them in the clinical practice. Further studies in larger independent cohort of specimens with different subtypes will help to determine the possible use of BORA as a biomarker in OC.

The poor survival seen in OC patients that harbor high BORA levels might be related to the ability of BORA to enhance the oncogenic capabilities of these tumors. In fact, this is, to the best of our knowledge, the first report that shows that high levels of BORA drive malignant cellular transformation in nontumoral cells in vitro. However, our results in vivo led us to hypothesize that BORA might be necessary (as we observed an initial attempt for engraftment), but not sufficient to promote tumorigenesis, and that other oncogenic hits presumably accompany BORA overexpression for OC cell transformation, as happens in tumoral SK-OV-3 line. Remarkably, SK-OV-3 cells harbor mutations in TP53, PI3KCA, ATM and ARID1A, already described as oncogenic events in OC [12], and mutations in NOTCH2, FBXW7 and APC not yet linked to OC tumorigenesis. It is possible that BORA oncogenic properties are mediated by increased PLK1 activity, which has been already described to be able to transform 3T3 NIH fibroblasts [46] and epithelial nontumoral prostate cells [47]. However, we are exploring, whether BORA in the oncogenic context of OC interacts also with other proteins, hitherto unknown, to drive malignant transformation.

In addition to the described oncogenic functions of BORA and its prognostic value in ovarian tumors, our data also highlights BORA as a potential novel therapeutic target. BORA displays an essential role in maintaining the proliferation and viability of OC cells. Contrary to what occurs with the nontumoral IOSE cell line, BORA silencing impairs cellular growth by triggering G2/M phase arrest and a subsequent caspase-dependent apoptosis in OC lines. Our transcriptomic analysis defined BORA as essential regulator of proliferation, viability and migration in OC. Supported also by the fact that suppression of BORA levels in vivo impairs tumor progression, BORA silencing represents an advantage when aiming at specifically targeting malignant dividing cells. On the other hand, as most OC-related deaths are associated with peritoneal tumor spread, we established a preclinical ex vivo model using patient-derived ascitic cells from advanced OC patients to test the effects of BORA depletion. In these cells, the reduction of BORA levels caused a decrease in the sphere forming capacities and cell viability, thereby indicating that BORA is still playing a relevant role in advanced stages of the disease. Further studies are required to evaluate potential synergies with front-line chemotherapy or other targeted therapies and also to molecularly identify those subsets of patients that could maximally benefit of BORA inhibition. PLK1 inhibition is lethal in *K-RAS* mutant cells as described in lung cancer [48], and in OC, synthetic lethality is produced with PLK1 inhibition and paclitaxel in *CCNE1* amplified HGSC cells [49]. Thus, it will be interesting in the future to search for hits that together with lack of BORA (or PLK1) will result into synthetic lethality in OC.

While this study focused primarily on BORA, our transcriptomic data uncovered multiple pathways previously linked to activated-PLK1 functions. For instance, a gene expression microarray analysis of PLK1 knockdown in bladder cancer cells [50] revealed GO biological processes, such as cell cycle, focal adhesions, VEGF and NF-κB signaling pathways deregulated, similar to what we found in our transcriptomic analysis, hence underlying most of the pathways modulated by BORA are PLK1-dependent. Increasing evidence also supports that PLK1 has multiple nonmitotic functions in cancerous and noncancerous cells [51]. Intriguingly, we found that BORA silencing modulates pathways related to cardiovascular homeostasis mainly due to the recent described function of PLK1 activity in regulation vascular smooth muscle cells [52]. Strikingly, these data evidence for the first time the participation of BORA activating PLK1 kinase to carry out nonmitotic functions. It will therefore be interesting to determine whether among all pathways described, any of them are PLK1-independent and modulated by BORA through other downstream effectors. Notwithstanding, the oncogenic phenotype upon BORA higher expression might be a result of the chromosome instability occasioned, but even in that case, we highlight that these results define BORA as a promising molecular target for the development of drugs that might reshape or improve the action of PLK1 inhibitors.

While different strategies are being considered by several groups to block BORA function (e.g., small-molecules, designed peptides to block the BORA-PLK1 axis and/or siRNA nanoencapsulation), another feasible alternative is to seek for actionable and druggable downstream effectors of BORA. Our transcriptomic analysis showed a significant modulation of genes involved in cellular viability and proliferation. Among them, BCL2 and CDK6 were found downregulated at mRNA levels and then confirmed at protein levels in primary patient-derived cancer cells and tumor lysates from mice upon BORA depletion. In fact, BH3 mimetics are emerging as promising avenues against cancer [53], particularly in OC, where BCL2 inhibition has proven to sensitize OC cells to DNA damage agents [54]. While CDK6 is reported to protect OC from platinum-induced death and predicts poor survival [55], its inhibition synergizes with cisplatin and may cause fewer side effects in nontumoral cells [56]. Our approach of simultaneously targeting both pathways showed a potent in vitro and ex vivo cell growth suppression, thereby providing a proof-of-principle rational of managing OC. Concurrent to our results, de Dominicie et al. [57] recently reported that pharmacologic inhibition of CDK6 and BCL2 markedly suppressed ex vivo and in vivo viability of leukemia cells.

In summary, the identification of mitotic actionable candidates by our integrative approach encourages us to further functionally characterize their implication in OC as potential anticancer targets. In particular, our experimental approach unveils BORA as an oncogenic regulator in OC, which modulates multiple cancer related key processes and offers potential therapeutic avenues for the future, either by targeting BORA itself or its downstream effectors.

## 4. Material and Methods

### 4.1. Analysis of Cancer Genomic And Transcriptomic Datasets

Raw data files from GSE14407 data set were downloaded from Gene Expression Omnibus (GEO) and analyzed with the Transcriptome Analysis Console (TAC) Affymetrix software (3.0, Affymetrix, Waltham, MA, USA). Gene expression profiles of BORA in the four representative OC data sets comparing benign versus tumoral samples were analyzed by GEO2R tool. The TCGA dataset repositories were interrogated by the R2 genomics website. Expression data for clinical stage, histological grade and gene correlations were obtained from the ovarian TCGA cohort and plotted on GraphPad Prism Software (6.0, GraphPad Software, San Diego, CA, USA). For overall survival analysis, the Kaplan–Meier Plotter platform for breast, lung, liver and ovarian cancer was mined. Briefly, BORA Affymetrix ID (219544_at) was introduced in the platform and patients were split by the autoselect best cut off given by the online tool. The autoselect best cut off computes all possible cut-off values between the lower and upper quartiles and selects the best performing threshold as a cut off. Hazard ratio, LogRank *p*-value and False Discovery Rate (FDR) are indicated in each graph for each tumor. The number of split patients in each group (high vs. risk) is also indicated in the graph at each time-point. DAVID Bioinformatics was used for the identification of Kyoto Encyclopedia of Genes and Genomes (KEGG) and biological processes specifically enriched in set of pathways. All data sets, bioinformatics tools and techniques used are explained in detail and referenced in Appendix A.

### 4.2. Human Samples

Fresh tissue samples including benign lesions and ovarian primary tumors were obtained immediately after surgery and stored at −80°C until processing at Vall Hebron University Hospital (VHUH) (Appendix A). All tumors were examined by the pathologist to confirm the diagnosis. Thirteen patient-matched paired tumors and metastases formalin-fixed paraffin-embedded (FFPE) were obtained from VHUH (Appendix A). Patient derived ascites-fluid was collected the day of the surgery from advanced OC patients (Appendix A). Approval to collect OC specimens was complying to the Institutional Review Board (IRB): PRAMI3082015. All patients gave their written informed consent.

### 4.3. Cell Lines

OAW42, 59M, OAW28, OVCAR4 and TOVD112 cell lines were acquired from European Collection of Authenticated Cell Cultures (ECACC, Porton Down, England) whereas SK-OV-3, UWB1.289 and UWB1.289+BRCA1 were acquired from American Type Culture Collection (ATCC, Manassas, VA, USA). A2780p and A2780cis were a generous gift from Dr. Francesc Viñals (IDIBELL, L’Hospitalet de Llobregat, Spain). BIN-67 was also kindly provided by Dr. Barbara Vanderhyden (Ottawa Hospital Research Institute, Ottawa, ON, Canada) and IGROV-1 by Dr. Antonio Rosato (Istituto Oncologico Veneto, Padova, Italy). IOSE-503 and IOSE-385 cell lines were obtained from the Ovarian Cancer Research Team OvCaRe (Vancouver, BC, Canada). HEK-293T was acquired from Dr. Erich A. Nigg’s lab (Universitat Basel, Basel, Switzerland). Cells were cultured the indicated culture media (Appendix A), regularly tested for mycoplasma contamination and stored in liquid nitrogen. Primary cells from passages 2–6 were used for the analysis. All cells were maintained at 37 °C in a saturated atmosphere of 95% air and 5% CO_2_. 

### 4.4. Protein Extraction, Immunoblot And Densiometry

Protein extracts were obtained in 1× RIPA buffer (Tris HCl 1.5 M Ph = 8.8, 318 NaCl 5 M, Triton X-100, EDTA 500 mM) supplemented with 1× EDTA-free complete protease inhibitor (Roche, Indianapolis, IN, USA) and phosphatase cocktail inhibitors (P5726, P0044, Sigma, St. Louis, MO, USA). Proteins (50–100 μg) were resolved in NuPAGE 4–12% Bis-Tris gels and transferred onto polyvinylidene difluoride (PVDF) membranes blocked for 1h with 5% nonfat milk (Panreac, Castellar del Vallés, Spain) or 5% bovine serum albumin (Sigma) and probed overnight at 4 °C with the indicated antibodies (Appendix A). Densitometry of each blot is represented in the panels within the figures. Densitometry was carried out using the Image J software. Each protein expression was compared to β-actin or Tubulin and referred to its control condition.

### 4.5. Real-Time Quantitative PCR

Total RNA was extracted from both human samples and cell lysates using RNeasy Mini Kit (Qiagen, Hilden, Germany). Two to three cubic millimeters of fresh-frozen tissues were homogenized using FastPrep-24 Lysing Matrix tubes (MP Biomedicals, Irvine, CA, USA) and 700 μL of QIAzol reagent, with Fast-Prep-24 Classic instrument (116004500 MP Biomedicals) (30 sec at 6.5 r/p for three times). A volume of 140 μL of chloroform was then added to the homogenates to continue with the extraction. For FFPE samples FFPE miRNeasy Mini Kit (Qiagen) was used. 1 μg of RNA was subjected to DNAse treatment and retrotranscription using SuperScript III (Qiagen). Real time PCR of BORA and GAPDH genes were performed using Taqman probes (Hs00227229_m1 for BORA and Hs02786624_g1 for GAPDH). For BCL2, CDK6, RERG, CLASP2, MARK2, SFRP1, SLC25A10, MMP7, IL1B, TPM1, MAD2L1, SHROOM2, RHOB and GAPDH genes, RT-qPCR was performed using SYBR green fluorescence (Applied Biosystems, Foster City, CA, USA). Primer sequences are listed in Appendix A. Relative quantification of gene expression was performed with the 2^(−ΔΔCt)^ method [58].

### 4.6. Plasmids,Lentiviral Production and Generation of Stable Lines

For stable depletion of BORA, short hairpin RNAs targeting the coding sequence 5′-CCGGTTGATAATGGCAGTTTA-3′ for shBORA#1 and 5′-TAACTAGTCCTTCGCCTATTT-3′ for shBORA#2 were designed and cloned into a pLKO.1-puro plasmid (Addgene Plasmid #10878). Control included nontargeting shRNA (shCTL) was purchased from Sigma Aldrich. pTRIPZ inducible lentiviral shRNAs were purchased from Open Biosystems (Dharmacon, GE Healthcare, Lafayette, CO, USA) (pTRIPZ_BORA-V1: V2THS_157921; pTRIPZ_BORA-V2: V3THS_393909). For BORA overexpression, the coding sequence of BORA cloned into pENTR/D-TOPO vector was obtained from Dr. Erich Nigg’s lab. Gateway LR Clonase II reaction (Life Technologies, Carlsbad, CA, USA) was used to transferred the BORA coding sequence into the pINDUCER20 lentiviral **system** (Addgene plasmid #44012), following the manufacturer’s protocol. Lentiviral particles were produced in HEK-293T cells as previously described [59]. Cells were transduced with fresh viral supernatant plus 5–8 μg/mL of polybrene. For selection of stable SK-OV-3 pTRIPZ lines, 0,75 μg/mL puromycin was added to the growth medium for five to seven days. IOSE-503 and SK-OV-3 pINDUCER20 transduced cell lines were selected with 0,5-1 mg/mL of G418 (Gibco, Billings, MT, USA) for five days. Stable cell lines were established from a mixed population of multiple clones to avoid clonal variation. BORA knockdown or overexpression was monitored by immunoblot or RT-qPCR.

### 4.7. Proliferation and Clonal Cell Growth Assays

For BORA silencing experiments, cells were seeded at 3 × 10^5^–1 × 10^6^ cells per p60 plate and infected with shCTL, shBORA#1 or shBORA#2 lentiviruses. For the proliferation experiments, 24 h post-infection cells were seeded at 2.5 × 10^3^–1 × 10^4^ cells per well on a 96-well plate. (*n* = 6/condition). At the indicated time points, cells were fixed in formaldehyde 4% solution and stored in phosphate-buffered saline at 4 °C. At the end of the experiment, cells were stained with 0.5% crystal violet. Crystals were dissolved with 15% acetic acid and optical density was read at 590 nm. Colony formation assays were performed by seeding SK-OV-3, A2780, IOSE-503 (5 × 10^2^–1 × 10^3^ cells) onto six-well plates in triplicates. Media was refreshed every three days and in the indicated wells doxycycline was added. After 9–11 days, cells were stained with 0.5% crystal violet, photographed and scored.

### 4.8. Anchorage Independent Growth Assay (Soft Agar)

To monitor anchorage independent growth, IOSE-503 and SK-OV-3 transduced cell lines (pIND_EV or pIND_BORA) were suspended in complete mixt medium containing 0.3% agar with or without 0.25 μg/mL doxycycline and then plated onto six-well plates on top of 0.6% agar in mixt medium previously polymerized (n = 3/condition). Cultures were grown for 21 days until colonies were visible. Macroscopically colonies were photographed (three fields per well) and scored. Differences in colony formation were assessed by comparing the number of colonies against control (pIND_EV).

### 4.9. Boyden Chamber Migration Assay

Transparent PET membrane 8.0 μm pore size inserts for 24-well plates (Falcon, Corning, Tewksbury, MA, USA) were used. Cells were loaded in the upper compartment of the cell culture inserts, while 600 μL of medium with 10% FBS were added in the lower compartment. Cells were allowed to migrate for four hours at 37 °C and then fixed with 4% formaldehyde during 30 min at RT. Cells that had attached to the membrane but not migrated were completely removed using a cotton swab. Migrated cells were stained and counted with 1 μL/mL Hoechst 33258 (Sigma-Aldrich, St. Louis, MO, USA) for 10 min at RT.

### 4.10. Growth Inhibition Activity Assay

Cell density is a signal for inhibition of cell growth. To monitor whether the overexpression of BORA could bypass this inhibitory signal, IOSE-503 pIND_EV and pIND_BORA cells were seeded onto six-well plates (3 × 10^5^ cells/well; i.e., high density) and incubated at 37 °C in a humidified atmosphere at 5% CO_2_ during 10-14 days after 100% of confluence. Cell medium was refreshed each three to four days and plates were photographed every three to four days.

### 4.11. Fluorescence-Activated Cell Sorter Analysis (FACs)

Cell lysates were fixed by adding 700 µL of ice-cold absolute EtOH. Cells were then pelleted at 5.000 rpm, washed twice with PBS and resuspended in 1 mL of PBS containing 1.14 mM sodium citrate (Sigma-Aldrich), 15 µg/mL propidium iodide (Sigma-Aldrich) and 300 µg/mL RNAse A (Panreac, Castellar del Vallés, Spain). Samples were incubated overnight at 4 °C. Data acquisition was performed with a FACS Calibur analyzer (BD Biosciences, Allschwil, Switzerland). Cell aggregates were excluded using pulse processing and a minimum of 10,000 events were measured per sample. Cell cycle phase distribution was analyzed using the FlowJo 9.6.4 software (Treestar, Woodburn, OR, USA).

### 4.12. Cell Death Assay

Cells transduced with either shCTL or shBORA#2 were plated onto 24-well plates (15 × 10^3^–10 × 10^4^ cells/well). Ninety-six hours post transduction, cells were stained with 0.05 mg/mL Hoechst for 30 min at RT. Condensed or fragmented nuclei were counted as dead cells as described previously [43]. 

### 4.13. Mouse Xenograft

All animal experimental procedures were approved by the Vall d’Hebron Hospital Animal Experimentation Ethics Committee (protocol number 04/18). Animal care was in accordance with VHIR guidelines. For the overexpression of BORA, 5 × 10^5^ or 5 × 10^6^ stable IOSE pIND_EV and IOSE pIND_BORA cells were injected into the flank of seven-week old female nude NMRI mice (ENVIGO, Indianapolis, IN, USA; *n* = 7 mice/group) in 300 μL of PBS 1X and Matrigel (1:1). SK-OV-3 explanted cells transduced with pIND_EV or pIND_BORA were injected in seven-week old female NMRI nude mice (ENVIGO; 2 × 10^6^ cells; *n* = 7 mice/group) in 300 μL of PBS 1× and Matrigel (1:1). Doxycycline (1 mg/mL) was added to the drinking water with 2% sucrose (Fisher-Scientific, Hampton, NH, USA) one-week pre-subcutaneous injection and during the whole experiment. For the constitutive depletion of BORA, 2 × 10^6^ cells of SK-OV-3 explanted cells transduced with shCTL or shBORA#2 lentiviral particles for 48 h were injected into the flank of seven-week old female NMRI nude mice (ENVIGO; *n* = 7 mice/group) in 300 μL of PBS 1× and Matrigel (1:1). For the inducible depletion of BORA, 2 × 10^6^ of SK-OV-3 pTRIPZ_BORA_V1 cells were injected subcutaneously into the flank of seven-week old female NMRI nude mice (ENVIGO; *n* = 14 mice). When tumors reached the volume of 150 mm^3^ on average, mice were randomly distributed into two experimental groups (*n* = 7 mice/group): one group was maintained with the vehicle (2% of sucrose) and a second group was treated with doxycycline-supplemented water (1 mg/mL) with 2% of sucrose ad libitum. Tumor engraftment and volume was monitored every two to three days using an electronic caliper. At end point, mice were euthanized and tumors were removed and weighted. Tumors were fixed in 10% formalin paraffin embedded for H&E and Ki67 staining or snap frozen in liquid nitrogen and stored at −80 °C.

### 4.14. 3D Culture from Patient-Derived Ascitic Cells

Tumoral cells from patient-derived ascitic fluid were isolated and cultured in anchorage independent conditions as previously described [42,43]. Briefly, tumoral cells were grown in serum-free 1:1-MCDB105:M199 media supplemented with 2mM L-Glutamine (Invitrogen, Carlsbad, CA, USA), B27 vitamin (Invitrogen Carlsbad, CA, USA), 20 ng/mL EGF (ProSpec-Tany Technogene Ltd, Rehovot, Israel) and 20 ng/mL FGF (ProSpec-Tany Technogene Ltd). For spheroid scoring, 15 × 10^3^ cells were reversely transduced and seeded in nonadherent 24-well plates (coated with 0.5% agar in nonsupplemented 1:1-MCDB105:M199 media). For MTS assay and protein collection, 3 × 10^5^ cells were seeded in nonadherent 6-well plate. At 96 h, spheres were collected for viability assay and protein extraction. Tumor spheroids were disaggregated with 0.5 mL of 1× StemPro® Accutase® (ThermoFisher, Waltham, MA, USA) and PMS:MTS (1:20) mixture was added 1:10 to each well containing 100 μL of disaggregated cells, and optical density was measured at 2–5 h.

### 4.15. Microarray Gene Expression Analysis

Transcriptome expression profiling of triplicate experimental samples for SK-OV-3 transduced with shCTL or shBORA cells was performed using the Affymetrix microarray platform with the Human Clariom^TM^ S assay. The quality and concentration of RNA was measured by RNA Nano Chip Bioanalyzer (Agilent Technologies, Santa Clara, CA, USA). First and second strand cDNA production, biotinylation, hybridization, labeling and scanning of the chips were performed at the VHIR Genomic facility. CEL files are publicly available at the GEOarchive repository GSE133635.

### 4.16. Drug Combination Studies

SK-OV-3 and A2780p cells seeded onto 96-well plates were treated with Palbociclib (Selleckchem, Houston, TX, USA), Abemaciclib (Selleckchem), Navitoclax (APExBio, Houston, TX, USA) and Venetoclax (APExBio, Houston, TX, USA) in a range of 0.01 to 25 µM or the combination of Palbociclib and Navitoclax. After five days, cells were fixed and stained with crystal violet. Combination index analysis was determined by the Chou–Talalay method using the Compusyn Software (1.0, ComboSyn Inc., Paramus, NJ, USA). For tumor patient-derived spheroids, spheres were incubated with the indicated drugs and their viability was assessed by MTS assay and immunoblot after 96 h.

### 4.17. Immunohistochemistry

Tumor tissues collected from the mouse were fixed in formalin 10% for two days and then stored in PBS until they were included in paraffin in the Pathology Department from the Vall Hebron Hospital. Immunohistochemistry was performed for Ki67 detection on a BenchMark Ultra platform (Ventana Medical Systems, Oro Valley, AZ, USA) following the recommended protocol. Antigen retrieval was performed with CC1 buffer for 20 min. A rabbit monoclonal 30-9 antibody was used to detect Ki67 protein (Ventana Medical Systems) and the incubation time was of 32 min. Detection system was Ultraview DAB, used following the recommended protocol. The percentage of tumoral and Ki67 positive cells was evaluated attending the help of a pathologist, considering the number of tumor cells in three high power fields detected morphologically and counting the number of nuclear stained cells with Ki67 immunohistochemistry using the ImageJ software. Hematoxylin and eosin (H&E) staining was performed following conventional procedures.

### 4.18. Statistical Methodologies

Unless otherwise indicated, mean ± SEM values are representative of three independent experiments. Statistical significance was determined using Prism 6 (6.0, GraphPad Software, San Diego, CA, USA). All statistical tests of comparative data were done using two-sided, unpaired Student´s *t*-tests, One- or Two-way analysis of variance (ANOVA). Correlation analysis was performed using Pearson´s test or, alternatively, Spearman’s test for nonnormal distributions. Data with *p*-value < 0.05 were considered statistically significant: * *p* < 0.05; ** *p* < 0.01; *** *p* < 0.001; **** *p* <0.0001.

## 5. Conclusions

Clinical management of ovarian cancer remains a challenge owing to the failure to obtain long-lasting benefits and the development of resistance to current standard therapies. Since the mitotic spindle is a validated therapeutic target against cancer, using an integrative global transcriptional profiling, we searched for novel mitotic target candidates, focusing our attention on BORA. Our results provide the first evidence of unanticipated oncogenic functions of BORA in addition to its previously described role in mitosis. Our data pinpoint BORA as a prognostic biomarker and as an essential mediator of tumor cell proliferation and migration in OC. BORA ablation attenuated tumor growth in vivo and compromised the viability of patient-derived tumor cells ex vivo, rendering it a potential therapeutic target. Furthermore, with the exploration of the BORA silencing transcriptional landscape, we identified downstream effectors such as CDK6 and BCL2 whose inhibition can be used as targeted therapies for ovarian cancer management.

## Figures and Tables

**Figure 1 cancers-12-00886-f001:**
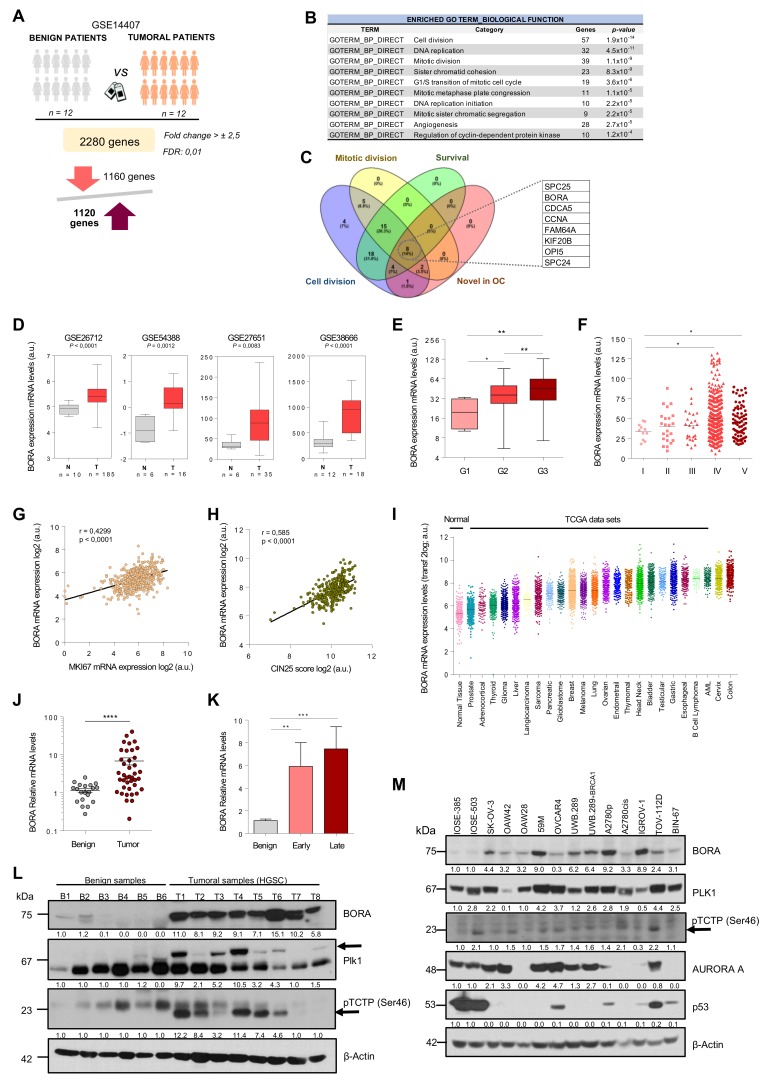
**BORA is overexpressed in OC associated with poor prognosis.** (**A**) Gene expression analysis comparing human epithelial ovarian surface tissues to ovarian carcinoma ovaries using data contained in the GSE 14407 dataset. (**B**) Functional annotation of upregulated genes reported by DAVID Bioinformatics 6.8. (**C**) Venn diagram illustrating the overlap of genes with the indicated filters. (**D**) Examination of BORA mRNA expression in published ovarian transcriptomic profiles comparing benign versus tumoral samples. (**E**,**F**) BORA mRNA expression levels in TCGA ovarian samples (*n* = 541 specimens) categorized by histological grade and clinical disease stage. (**G**) Correlation (Pearson) between BORA mRNA levels to MKi67 transcriptional expression and to (**H**) CIN25 expression in the ovarian TCGA cohort. (**I**) BORA mRNA expression levels in different tumor types from the TCGA repository. Data were retrieved from the TCGA database using the R2 Genomic Visualization platform. (**J**) BORA expression is higher in tumoral OC primary samples (*n* = 40) compared to benign ovaries (*n* = 20) and (**K**) correlates with late clinical stage. All values of mRNA were normalized to *GAPDH*. (**L**) Immunoblot showing protein levels of BORA, PLK1 and pTCTP-Ser46 in fresh-tissues from benign (*n* = 6) and HGSC primary ovaries (*n* = 8). (**M**) Immunoblot analysis of BORA, PLK1, pTCTP-Ser46, AURORA A and p53 in ovarian cell lines, including 12 tumoral lines and two nontumoral IOSE clones. β-Actin was used to control equal protein loading in L and M. *P*-values were calculated using a two-tailed Student’s *t*-test. * *p* < 0.05; ** *p* < 0.01; *** *p* < 0.001, **** *p* < 0.0001.

**Figure 2 cancers-12-00886-f002:**
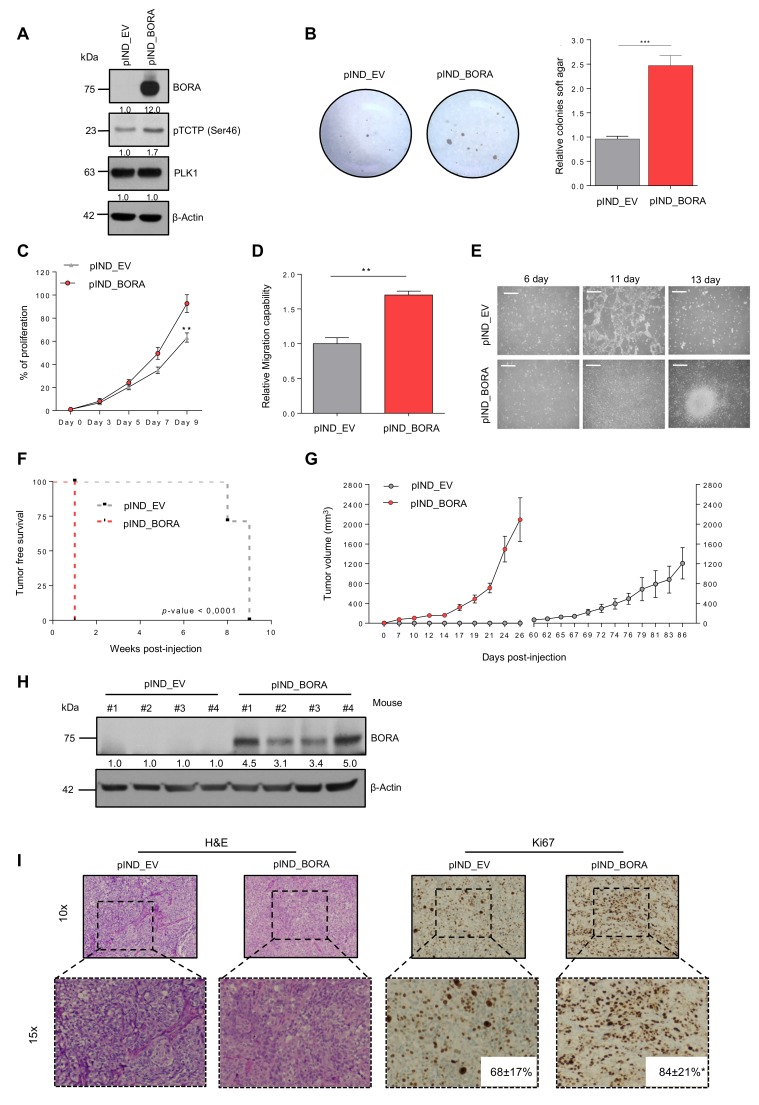
**BORA contributes to the transformation of epithelial ovarian cells in vitro and favors malignant features in vivo.** (**A**) Immunoblot showing BORA, pTCTP-Ser46 and PLK1 levels in IOSE-pIND_EV and IOSE-pIND_BORA lines upon doxycycline administration (0.25 μg/mL). (**B**) Representative macroscopic images of anchorage independent colony formation growth in soft agar and average quantification of three independent experiments ± SEM. (**C**,**D**) Normalized proliferation rates and migration capacities and of IOSE-pIND_EV and IOSE-pIND_BORA cell lines. Graphs represent an average of three independent experiment ± SEM. *P*-value was calculated using a two-tailed Studen’s *t*-test. ** *p* < 0.01. (**E**) Images showing the loss of contact inhibition capacities upon BORA overexpression in the IOSE cell line after reaching confluence. Bar: 100 μm. (**F**) Percentage of SK-OV-3 tumors developed (incidence) in each group (*n* = 7 mice/group) and the required days from cell injection until the tumors were measurable (latency). *P*-values were estimated using a log-rank test to determine the difference in appearance between pIND_EV tumors (grey line) versus pIND_BORA tumors (red line). (**G**) Tumor volume measured every two to three days. Error bars represent SEM. (**H**) Immunoblot analysis of BORA levels in representative xenograft tumors from both experimental groups. β-Actin was used as loading control. (**I**) Representative microscopic H&E and Ki67–stained images from the OC xenografts. Ki67 percentage presented as mean and standard deviation are included in the images for both groups. *P*-value was calculated using a two-tailed Student’s *t*-test. *** *p* < 0.001.

**Figure 3 cancers-12-00886-f003:**
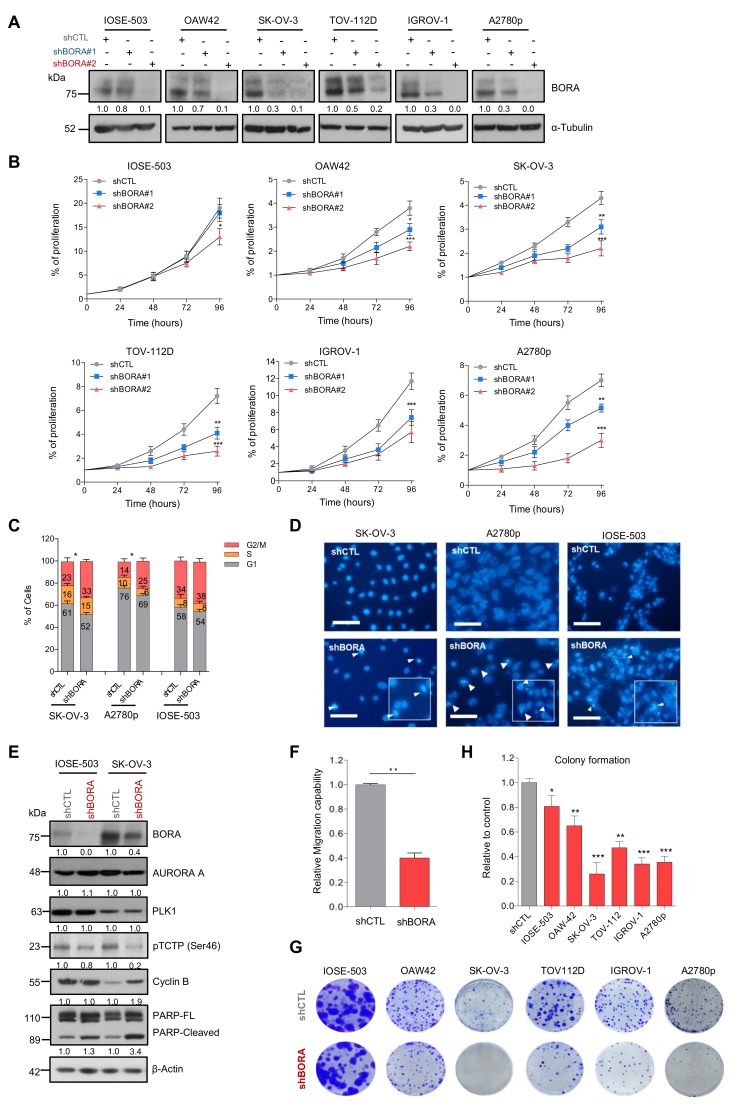
BORA silencing attenuates proliferation, increases the number of cells in G2/M phase and induces apoptosis cell death in vitro. (**A**) BORA levels detected by immunoblot in the indicated cell lines infected with nonsilencing control (shCTL) or two different shRNA against BORA (shBORA#1 and shBORA#2). α-Tubulin was used as loading control. (**B**) Normalized proliferation curves of the indicated OC cell lines infected with shCTL, shBORA#1 and shBORA#2 (herein shBORA), measured by crystal violet staining. (**C**) Cell cycle profile of SK-OV-3, A2780p and IOSE-503 cell lines infected with shCTL and shBORA viruses. (**D**) Hoechst staining images of shCTL or shBORA-infected cells in the three lines. Arrowheads point at nuclei with condensed or fragmented chromatin. Bar: 100 μm. (**E**) Representative immunoblot of BORA, PLK1, AURORA A, pTCTP-Ser46 cyclin B1 and PARP proteins upon BORA silencing in the SK-OV-3 and IOSE-503 cell line. β-Actin was used as loading control. (**F**) Migration capacities of shCTL and shBORA-SK-OV-3 infected cells. (**G**) Representative macroscopic images of colony formation assay in shCTL and shBORA- transduced ovarian cell lines after 10–12 days of culture and (**H**) average quantification of three independent experiments. Graphs represent an average of three independent experiments ± SEM. For B, C, F and H, *P*-value was calculated using a two-tailed Student’s *t*- test. * *p* < 0.05; ** *p*< 0.01; *** *p* < 0.001.

**Figure 4 cancers-12-00886-f004:**
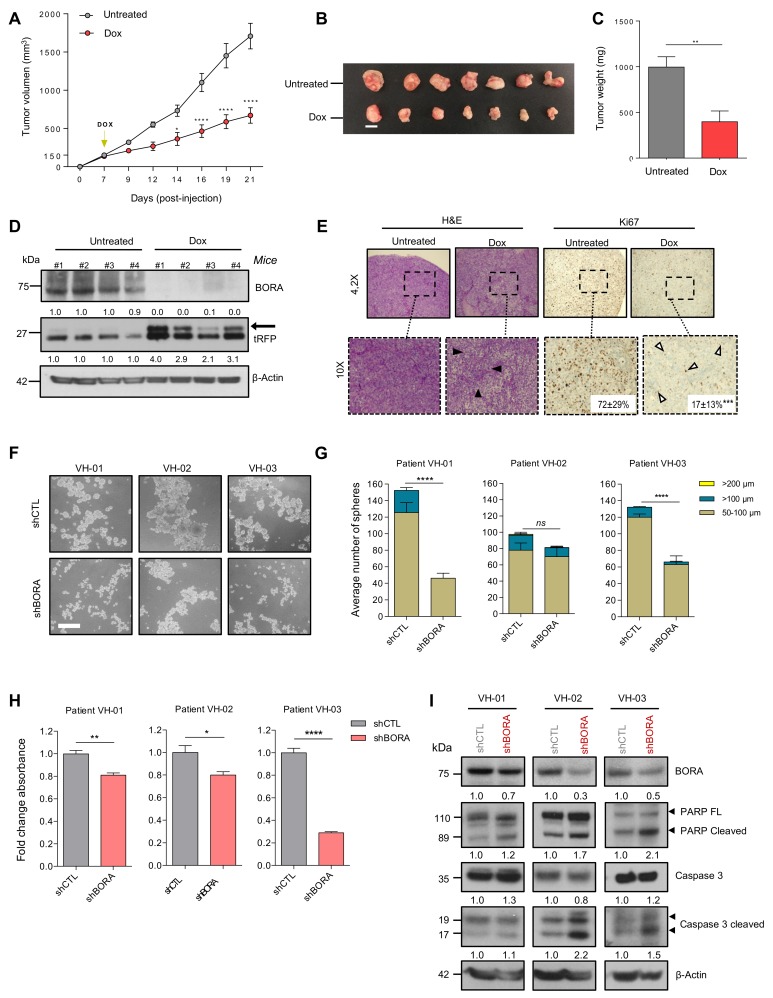
**BORA ablation impairs growth of tumor cells in vivo and viability of patient- derived spheroids ex vivo.** (**A**) Tumor growth of SK-OV-3 pTRIPZ_BORAV1 xenotransplant transduced cells untreated (grey line) or after a sustained treatment with doxycycline supplemented in drinking water (red line) (*n* = 7 mice/group). Two-way analysis of variance (ANOVA) was used to calculate the significance (*p*-value) of the difference between the untreated and doxycycline treated group. * *p* < 0.05; **** *p* <0.0001. (**B**) Representative tumors collected at the time of euthanasia. Bar: 1 cm. (**C**) Average weight of the tumors. (**D**) Immunoblot analysis of BORA and tRFP protein levels in 4 representative xenograft tumors from both experimental groups. β-Actin was used as loading control. (**E**) Immunohistochemical analysis of H&E and Ki67 staining. Representative microscopic stained-images of the OC xenografts from the doxycycline-untreated and treated groups. Ki67 percentage as mean and standard deviation are included in the images for both groups (**F**) Representative images of the three OC patient-derived ascitic cells grown under anchorage independent conditions with shCTL and shBORA lentiviral transduction. Scale bar: 100 μm. (**G**) The number of spheres was scored 36 h post-transduction with shCTL and shBORA in the indicated ascitic primary cells, classified according to the diameter (50–100 μm, ≥100 μm and ≥200 μm). (**H**) Viability assay (MTS) was performed at 96 h post-transduction with shCTL or shBORA in the indicated OC patient-derived ascitic cells. (**I**) Spheres were used for protein extraction and immunoblot analysis with the indicated apoptosis antibodies 96 h post-transduction. Graphs represent an average of three independent experiments ± SEM. β-Actin was used as loading control. For figure C, E, G and H *P*-values were calculated using a two-tailed Student’s *t*-test. * *p* < 0.05; ** *p* < 0.01; **** *p* < 0.0001.

**Figure 5 cancers-12-00886-f005:**
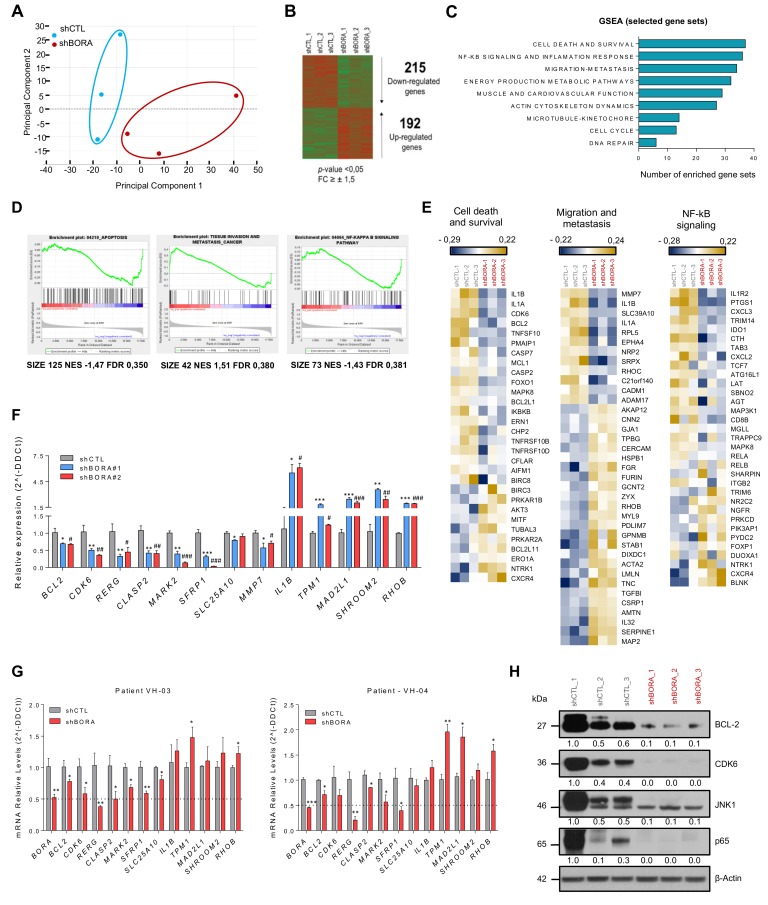
**BORA modulates the expression of survival, migration and NF-κB signaling genes.** (**A**) Principal component analysis illustrates segregation of distinct expression profiles of shCTL and shBORA groups (*n* = 3/group). Note that Principal Component 1, capturing around 38% of gene expression variance, effectively distinguishes BORA depleted and nondepleted samples. (**B**) Heat map comparing the transcriptional profiles of shCTL and shBORA samples. (**C**) Gene set enrichment analysis using the GSEA software using the transcriptome data grouped in different term categories. Major collections from hallmarks, KEGG pathways and GO gene sets were downloaded from MSigDB.v6 and used for the GSEA. Graph represents total number of enriched gene-sets with FDR < 0.5 or *p*-value < 0.05 plotted regarding each category. (**D**) Representative GSEA curves for significant enriched gene sets related to cell death, migration and NF-κB signaling pathways. Corresponding size of the gene set, normalized enriched score (NES) and FDR for enriched gene sets are included. (**E**) Heat maps indicating selected differentially deregulated genes grouped in categories of the relevant pathways. The color key shows relative expression levels of the differentially expressed genes (yellow corresponds to overexpressed genes while blue corresponds to underexpressed genes). (**F**) Array validation by quantitative real-time PCR using two independent shRNA particles in the SK-OV-3 cell line. Values of shBORA#1 and shBORA#2 are represented as fold change versus shCTL. (**G**) Quantitative real-time PCR of the indicated genes in two patient-derived ascites cells transduced with either shCTL or shBORA viruses. *GAPDH* was used as endogenous control. Relative fold-change in expression was determined by the comparative 2(-ΔΔCt) method. (**H**) Immunoblot of the indicated proteins to verify the alteration in abovementioned pathways. Protein lysates used derived from the BORA and control depleted tumors from the in vivo experiment. β-Actin was used as a loading control. *p*-value was calculated in F and G using a two-tailed Student’s *t*- test. * Compares shCTL versus shBORA#1 and # compares shCTL versus shBORA#2; *, # *p*< 0.05; **, ## *p*< 0.01; ***, ### *p* < 0.001.

**Figure 6 cancers-12-00886-f006:**
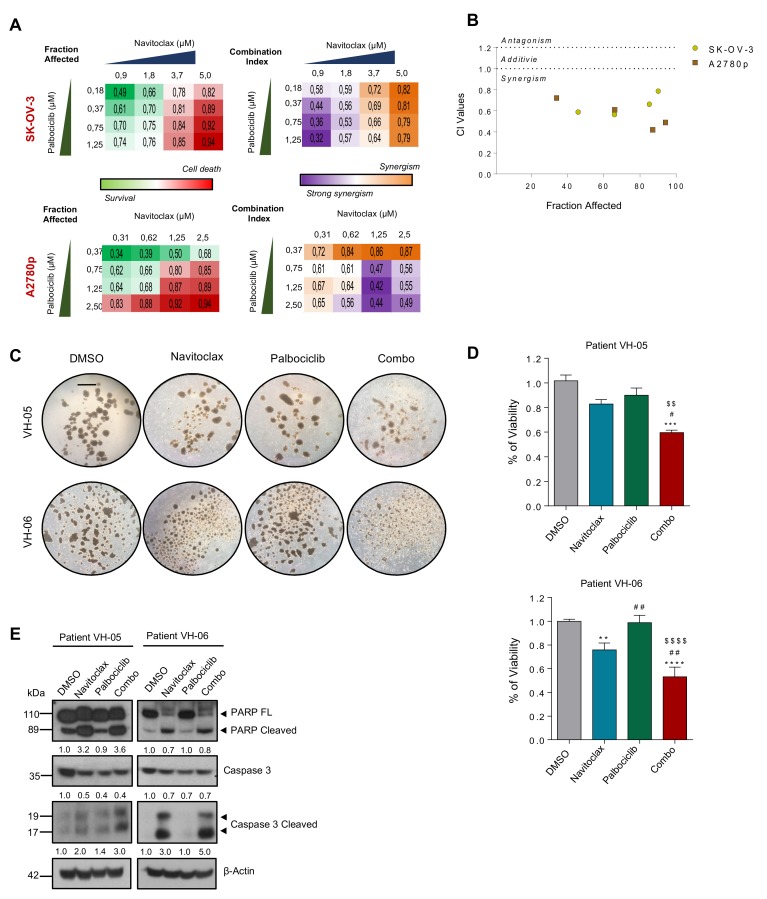
BCL2 and CDK6 inhibitors cooperate to reduce the viability of patient-derived ascitic cells grown in 3D. (**A**) Clustering results showing fraction affected (FA) and combination index (CI) of the top combinations evaluated in SK-OV-3 and A2780p cell line using nonconstant ratio with different concentrations of Navitoclax and Palbociclib. (**B**) Combination index (CI) of the indicated drugs. CI was calculated by the Chou–Talay method. Data plotted are CI value of both cell lines at different FA. (**C**) Representative images of two patient-derived tumoral cells grown under anchorage independent conditions (tumor spheroids) and treated with Palbociclib (12 μM) and Navitoclax (4 μM) during 96 h. Bar: 100 μm (**D**) Viability assay (MTS) was performed on the spheres treated during 96h of the indicated drugs and the combo. Values are represented as fold change versus the control (DMSO). *P*-values were calculated using a one-way ANOVA analysis. * compares DMSO versu*s* the rest of the conditions; # Navitoclax *versus* rest of conditions; $ Palbociclib versus Combo. # *p* < 0.05; **, ##, $$ *p* < 0.01; ***, ### *p* < 0.01; ****, $$$$ *p* < 0.001. (**E**) Immunoblot of the indicated protein markers to confirm the apoptosis upon simultaneous administration of the two inhibitors in primary patient-derived cells grown as spheres. β-Actin was used as a loading control.

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
