# Peer review of "Aurora Borealis (Bora), Which Promotes Plk1 Activation by Aurora A, Has an Oncogenic Role in Ovarian Cancer"

_cancers, 2020, doi:10.3390/cancers12040886_

Round 1

Reviewer 1 Report

This is a well-performed and interesting study investigating the role of BORA in ovarian cancer pathogenesis.

Major:

My primary criticism about this paper is that the SK-OV-3 and A2780 cell lines are now felt to be unlikely high grade serous ovarian cancer (Domcke et al, Nat Commun 2013, PMID: 23839242). However, the inclusion of human samples is a strength. The authors are strongly encouraged to use alternate cell lines that better represent high grade serous carcinoma for future studies.

Minor edits:

Lines 51-54 - Please add the citation for these statements about dataset analysis. If these lines are about results from the current manuscript, then please remove them from the introduction.

There are a number of minor grammatical errors, and the paper would benefit from a thorough read-though to address. For example:

Line 41 - should be changed to "is the responsibility"

Line 85 - should be changed to "is associated with"

Author Response

This is a well-performed and interesting study investigating the role of BORA in ovarian cancer pathogenesis. 

Major:

My primary criticism about this paper is that the SK-OV-3 and A2780 cell lines are now felt to be unlikely high grade serous ovarian cancer (Domcke et al, Nat Commun 2013, PMID: 23839242). However, the inclusion of human samples is a strength. The authors are strongly encouraged to use alternate cell lines that better represent high grade serous carcinoma for future studies. 

We acknowledge Reviewer 1 for his/her comment and agree that there is considerable controversy about the histopathological origin of the cell lines. As Domcke et al. pointed out, the most commonly used cell line models for ovarian cancer - and implicitly for the most prevalent subtype HGSOC – are SK-OV-3, A2780, OVCAR-3, CAOV3 and IGROV1 (quantified via Pubmed citations, as they mentioned). In fact, SK-OV-3 cells were obtained from a patient with ovarian cancer of serous histology (Hernandez L et al., Gynecol Oncol. 2016;142(2):332–340; original reference by Fogh J et al., Journal of the National Cancer Institute. 1977;58(2):209–14). However, Domcke and co-workers thoroughly showed how while OVCAR4, OAW28 and 59M (also used in our manuscript) are likely high-grade serous, SKOV3, OAW42 and A2780p are unlikely to behave as such.

We would like to emphasize that our manuscript does not aim at focusing at high-grade serous type, but importantly; the origin of several of our patient-derived ascites used through the manuscript were derived from patients with serous ovarian cancer (VH-2 to VH-5; patients’ features described in Suppl. Table 5).

In line with Revierwer’s 1 comment and to not add confusion on the histological origin of the different cell lines, we have removed the nomenclatures above the Western blot panel in Fig. 1M and have ensured that the text does not emphasize that our results apply only to serous tumors. We will also follow his/her recommendation for future studies and select representative cell lines when focusing specifically in HGSOC.

Minor edits:

Lines 51-54 - Please add the citation for these statements about dataset analysis. If these lines are about results from the current manuscript, then please remove them from the introduction.

Lines 51-54 describe results from our group (current manuscript). As suggested by the reviewer we have removed this information and add it to results description of our results later on (line 85 on) (see new version of the manuscript).

There are a number of minor grammatical errors, and the paper would benefit from a thorough read-though to address. For example:

Line 41 - should be changed to "is the responsibility"

Line 85 - should be changed to "is associated with"

We thank the reviewer for pointing out these grammatical errors. We have corrected these and other mistakes thorough the manuscript.

Reviewer 2 Report

The authors reported their findings on the role of BORA as an oncogenic factor in ovarian cancer. This study revealed the important pro-tumor activity of BORA and the therapeutic potential of targeting BORA downstream pathways. Several points need to be addressed before it can be considered for publication.

  1. Many HGSCs originated from the fallopian tube epithelial cells. The authors only used OSE as the health control cells. As an important cell-or-origin, fallopian tube epithelial cells should also been evaluated for their expression levels of BORA.
  2. The figures are in low resolution. It is hard to see some important details. (This may be caused by the submission process.)
  3. In Figure 1D-1I, the authors need to clarify if the data in these graphs are at mRNA level or protein level.
  4. In Figure 1L, PLK1 blot was incomplete, bands were cut off, and loading controls were over exposed and not equal. Is pTCTP(Ser46) a good marker for BORA and PLK1 activation? For example, in Figure 2A when BORA was overexpressed at such a high level, pTCTP(Ser46) was only increased slightly. Can the authors evaluate the levels of p-PLK1(Thr210)? In Figure 1M, pTCTP blot were cut off too much.
  5. In figure 2C, was the proliferation assay in 2D or 3D culture? It needs clarification.
  6. Statistical analysis is missing for all the Ki67 staining analysis, and figure 3C, 5F and 5G.
  7. Figure 3D needs quantification data and Statistical analysis.
  8. Supplemental 3 legend was mislabeled. (f and g)
  9. How many mice were used in each group? This information was missing in all the in vivo experiments. It should be included in the figures and/or the legend.
  10. Page 21, line 658, change to 3D culture.

Author Response

The authors reported their findings on the role of BORA as an oncogenic factor in ovarian cancer. This study revealed the important pro-tumor activity of BORA and the therapeutic potential of targeting BORA downstream pathways. Several points need to be addressed before it can be considered for publication.

Many HGSCs originated from the fallopian tube epithelial cells. The authors only used OSE as the health control cells. As an important cell-or-origin, fallopian tube epithelial cells should also been evaluated for their expression levels of BORA.

We acknowledge the Reviewer’s comment and agree with this point. There are very few epithelial cell lines derived from the fallopian tube (FTSECs). To the best of our knowledge, the only commercially available ones are FT33 cells (Alison M. Karst, et al., 2011). Although we did not confirmed our results on this cell line, we did evaluate BORA mRNA levels also in 20 human benign ovaries (Fig. 1J) and BORA protein levels in another cohort of benign vs tumoral ovaries (Fig. 1L).

The figures are in low resolution. It is hard to see some important details. (This may be caused by the submission process.)

We have also noticed that the resolution of the Figures included in the manuscript is not optimal. However, we believe this has happened during editorial processing as the original Figures are of high resolution and they display with high quality. We will adress this issue with the editors to ensure maximal resolution of the Figures.

In Figure 1D-1I, the authors need to clarify if the data in these graphs are at mRNA level or protein level.

Graphs from figures 1D to 1I represent BORA transcriptional levels. This is now clarified in in the axis of the graphs and also in the Figure legend.

In Figure 1L, PLK1 blot was incomplete, bands were cut off, and loading controls were over exposed and not equal. Is pTCTP(Ser46) a good marker for BORA and PLK1 activation? For example, in Figure 2A when BORA was overexpressed at such a high level, pTCTP(Ser46) was only increased slightly. Can the authors evaluate the levels of p-PLK1(Thr210)? In Figure 1M, pTCTP blot were cut off too much.

We thank the Reviewer for his/her comment. Regarding Fig. 1L, we have changed the PLK1 and b-actin blots to less exposed ones (New Fig. 1L). We have also now included densitometry measurement for all Western blot panels. As for PLK1 activation marks, p-PLK1 (Thr210) has been used as readout for PLK1 activation; however, western blots with this antibody are proven challenging. As an alternative readout for PLK1 activation, pTCTP-Ser46 phosphorylation is also widely used. Cucchi et al., demonstrated that phosphorylation at Ser46 in TCTP correlates with PLK1 kinase activity in G2 and mitotic cells, when Plk1 is reported to be activated by BORA (Cucchi et al., Anticancer Res. 30, 4973–4986 (2010).

Finally, in relation to the fact that BORA overexpression levels did correlate with only moderate levels of pTCTP-Ser46, the current view is that there is no direct correlation between Bora and pTCTP (Ser46) levels as Plk1 activation functions as a bi-stable switch, meaning that minimal amounts of BORA are needed to switch on PLK1 as described by Bruinsma et al. (2014). Thus, once BORA activates PLK1, the limiting factor is not BORA levels, but PLK1 levels, which do correlate with pTCTP levels as described in Cucchi et al (2010).

In figure 2C, was the proliferation assay in 2D or 3D culture? It needs clarification.

This proliferation assay was performed in 2D cultures. We have now clarified this point also in the text.

Statistical analysis is missing for all the Ki67 staining analysis, and figure 3C, 5F and 5G.

We have incorporated in the text the statistical significance for all Ki67 staining analysis and the flow cytometry data (Fig. 3C). For Figures 5F and 5G, we have incorporated the statistical significance within the panels.

Figure 3D needs quantification data and Statistical analysis.

The quantification and statistical analysis was shown in Supplemental Figure 5a.

Supplemental 3 legend was mislabeled. (f and g)

We have corrected it.

How many mice were used in each group? This information was missing in all the in vivo experiments. It should be included in the figures and/or the legend.

The information requested by the Reviewer was included in the Methods section. For all in vivo experiments, 7 mice were used per group. We have also included now this information in the corresponding main and supplemental Figure legends.

Page 21, line 658, change to 3D culture.

Reviewer 3 Report

In this study the authors examined the role of BORA in ovarian cancer within the preclinical setting. The results of the study are of potential interest however, there are some modifications that are required before the manuscript can be accepted.

Major comments:

  • The figures throughout the manuscript are very difficult to visualise and read.
  • Results section:
    • Supplementary Figure 1: when examining the dysregulated genes how did the authors split the high vs low groups. This should be described in the methods section.
    • Figure 2H the BORA blot and b-actin blot do not match. B-actin blot contains 7 proteins.
    • Figure 2I: how was Ki67 counts determined/counted? This is not described in the figure legends or methods section.
    • Section 2.4: the wording of this section is unclear. Based on the paragraph the cell lines did not grow in the mice. However, figure 2F implies there is tumours measured?
    • Within the flow-cytometry how was the data analysed?
    • Figure 3: it would appear that there is a modest reduction in protein expression could the authors provide densitometry data to determine the percentage of reduction.
    • Figure 3E: difficult to interpret, densitometry plots would give robust data, based on the current western shown, I do not see changes in cyclin B and actin levels are inconsistent.
    • Figure 5 not all genes were dysregulated in the patient derived data, could the authors comment on this.
    • Palbociclib and abemaciclib are inhibitors that target both CDK4 and 6. Preclinical and clinical studies have demonstrated that ER and Rb are potential biomarkers for these drugs. Based on the data shown it is a big leap that combination of CDK4/6 and Bcl-2 inhibitors inhibit the BORA signalling pathway since no data is shown to validate this.

Minor comments:

  • Figures are not in order
  • Figure 4 legend does not describe the figures looks like a copy paste from a different legend.
  • Materials and Methods
    • Section 4.1 do not define the techniques used in this study.
    • Section 4.8 does not describe method for colony scoring.
  • Minor grammar and spelling throughout the manuscript

If the manuscript is to be accepted for publication the authors should address the comments and revise accordingly.

Round 2

Reviewer 2 Report

The manuscript was significantly improved. Four minor points for perfecting the figures are the following.

Figure 3C, add error bars.

Figure 2I, Ki67 percentage data should be presented as mean+/-SD.

Figure 4E, Ki67 percentage data should be presented as mean+/-SD.

Figure 4G, add error bars.

Author Response

The manuscript was significantly improved. Four minor points for perfecting the figures are the following:

Figure 3C, add error bars.

Error bars have been added to new Figure 3C.

Figure 2I, Ki67 percentage data should be presented as mean+/-SD. 

Figure 4E, Ki67 percentage data should be presented as mean+/-SD.

Ki67 percentage in Figure 2I and 4E are now presented as mean+/-SD, as requested.

Figure 4G, add error bars.

Error bars have been added to new Figure 4G.